# Polydopamine-assisted PDGF-BB immobilization on PLGA fibrous substrate enhances wound healing via regulating anti-inflammatory and cytokine secretion

**Xiao Yang, Peng Zhan, Xiuyan Wang, Qiushuang Zhang, Yi Zhang, Haitao Fan\*, Ranwei Li\*, Ming Zhang** \*

Department of Urology, Jilin University Second Hospital, Changchun, PR China

\* zhangming196312@163.com (MZ); ranwei1968@sina.com (RL); jlfanht@sina.com (HF)

**Data Availability Statement:** All relevant data are within the paper and its Supporting Information files.

## Abstract

Platelet-derived growth factor-bb (PDGF-BB) is a potent chemokine and mitogen for fibroblasts, keratinocytes, and vascular endothelium in the injured area, believed to be effective in wound healing. However, the short half-life of PDGF-BB and its rapid release from the wound surface limited its efficacy in *vivo* and *vitro*. To evaluate the wound healing effects of dorsal skin in SD rats with polydopamine-assisted immobilized PDGF-BB on PLGA nanofibrous substrate. First, the effects of pDA-coating and PDGF-BB immobilization on the morphology, compositions, and hydrophilicity of substrates were evaluated in details. Second, the wound healing effect of pDA/PLGA/PDGF-BB substrate was assessed in the dorsal skin of SD rats. Last, the cytokine analysis by ELISA method was employed to evaluate the advantages of pDA/PLGA/PDGF-BB substrate on anti-inflammatory, angiogenesis, and cellular proliferation. This method significantly improved the immobilization amount and stability of PDGF-BB on the substrate ($p<0.01$), further improved the hydrophilicity of substrates ($p<0.05$). Furthermore, the wound closure process was much more accelerated in the pDA/PLGA/PDGF-BB group ($p<0.05$). H&E and CD31 staining informed that the wound treated by pDA/PLGA/PDGF-BB substrate showed a high degree of regeneration and angiogenesis. The cytokine analysis showed that pDA significantly reduced the high level of inflammatory cytokines such as TNF-α ($p<0.05$). And the immobilized PDGF-BB significantly elevated the level of TGF-β and VEGF ($p<0.05$). The pDA/PLGA/PDGF-BB substrate showed great therapeutic effect on wound healing compared with other control groups via regulating anti-inflammatory, angiogenesis, and cellular proliferation. Absolutely, this report offered an available novel method for skin regeneration.

## Introduction

Full-thickness skin injury caused by burns, bedsores or diabetic foot ulcers is an urgent clinical problem to be solved. In order to repair the damaged tissue including epidermis and dermis,

**Funding:** M.Z. Program of Scientific Development of Jilin Province (20180101103JC) http://www. jlkjxm.com/. R.L. Program of Scientific Development of Jilin Province (20180101103JC) http://www.jlkjxm.com/. The funders had no role in study design, data collection and analysis, decision to publish, or preparation of the manuscript.

**Competing interests:** The authors have declared that no competing interests exist.

various methods have been performed in wound area. Self-skin-grafting, often used as a conventional therapy, is considered to cause secondary injury and scarring. And it is not suitable for the extensive wounds beyond 2% of total body surface area due to the limited sources [1]. The application of heterogenous and allogeneic skin-grafting are supposed to solve these difficulties. However, it is hard to overcome the safety problems of biologics caused by immune rejection or cytotoxic responses [2–5]. Benefited from the development of the tissue engineering, human bioengineered skin implantations attracts a great interest in these field, and shows lots of advantages to defeat the problems mentioned above [1–7].

Poly[(rac-lactide)-co-glycolide] (PLGA), a FDA-approved polyester, has been widely used in the field of regenerative medicine due to its hydrolytic degradation and good controllability of macro properties. And electrospinning is a simple technique for the efficient production of micro to nano fibers from polymer solutions with high productivity and low cost. It was believed that electrospun fibers had the potential to achieve high drug loading and the ability to sustain drug release. Shih-Feng Chou studied the loading and release of tenofovir (TFV), a hydrophilic small molecule drug, into fibers of polycaprolactone (PCL) and poly (D,L-lactic-co-glycolic) acid (PLGA). The result showed that Sustained release of TFV depended on PLGA content. The PLGA fibers has outstanding sustained-release ability [8]. Amy Contreras' group encapsulated a photosensitizer (PS) in PCL/PLGA fibers in the PS inert state, so that the antibacterial function would be activated on-demand via a visible light source [9]. This study successfully demonstrates the significant potential of PS-encapsulated electrospun fibers as photodynamically active biomaterial for antibiotic-free infection control. These studies fully demonstrated the advantages of PLGA electrospinning in drug sustained release.

A variety of positive factors such as antimicrobial materials [10–12], growth factors [13], and physical stimulation (electrical stimulation) [14] were used to functionalize the implants to achieve better therapeutic effect. Among these beneficial factors, growth factor has positive biological activity on the proliferation, migration and differentiation of cells, and plays an important role in skin regeneration. PDGF comprises a family of homo or heterodimeric growth factors including PDGF-AA, PDGF-AB, PDGF-BB, PDGF-CC, and PDGF-DD. In the injury area, platelet-derived growth factor-bb (PDGF-BB) is a potent chemokine and mitogen for fibroblasts, keratinocytes, and vascular endothelium. It also stimulates macrophages to produce and secrete growth factors such as TGF-β. Hence, the PDGF-BB is often believed to be effective in wound healing. Recombinant human PDGF-BB (Becaplermin) is currently the only FDA-approved growth factor for the treatment of chronic trauma and has been successfully used for debridement and healing of advanced diabetic acromegaly ulcers based on gel dosage form [15]. Robert reported a novel therapy strategy combining PDGF-BB and a CXCR4 antagonist (AMD3100) to accelerate diabetic wound closure [16]. Specifically, it's informed that the topical increase in PDGF-BB was beneficial to the homing/implantation of *in situ* progenitor cells into the wound. Not only in pharmacy, but also in biomaterial engineering, the PDGF-BB has been applied to develop novel skin graft. Xie's group created a bionic system that could be used as a scaffold to support wound healing, while simultaneously releasing VEGF and PDGF-BB in a way that mimicked different stages of angiogenesis and cell proliferation to promote wound healing [17]. A nanofibrous composite containing PDGF-BB-loaded nanoparticles was fabricated by electrospinning. Slow expelling of PDGF-BB improved the epithelium regeneration, collagen deposition, and functional tissue remodeling. With this design, an accelerated wound healing was achieved on a full thickness rat skin wound model. It was also informed that the PDGF-BB control releasing device was promising great potential in the treatment of skin regeneration.

However, the potency of PDGF-BB control releasing device was sometimes "uncontrollable" when applied in *vitro* and *vivo* due to the burst release in the wound site. Excessive local

concentration of PDGF-BB may cause some unexpected side effects. In order to overcome these disadvantages, a growing interest was arisen in the development of biomaterials involving the immobilization of growth factors, which would allow these artificial materials to regulate specific cellular functions. In Jin's report [18], epidermal growth factor (EGF) and other multiple epidermal induction factors were encapsulated in fibrous scaffold by core–shell electrospinning. No burst release was occurred. And an enhanced ability for ADSCs epidermal differentiation was achieved [13]. In a recent work, Wang fabricated a chitin film containing basic fibroblast growth factor with a chitin-binding domain as wound dressing. This system showed great advantages in uptake and release of bFGF, and finally determined to be a promising and effective strategy for wound dressing as it could induce vascularization in *vivo* [19]. However, nether encapsulating the growth factor into the material nor recombinant protein was a convenient method for immobilization of growth factor.

Recently, a pDA-assisted immobilization strategy simulating mussel-adhesion phenomena in nature was utilized to strengthen the incorporation of peptides or growth factors on implants. And the enhanced regenerative capacity of the modified implants was also positively verified in *vitro* and *vivo*. Under alkaline conditions, a poly(dopamine) layer is formed on materials through the catechol groups. And the dopamine may offer a lot of activated amino and phenyl to immobilize peptides and proteins through hydrogen-bond interaction. Jeong Seok Lee's group fabricated a titanium (Ti) surface coated with pDA to facilitate the immobilization of BMP-2 [20]. The periodontal ligament stem cells (PDLSCs) cultured on pDA/BMP-2-Ti showed the highest osteogenic activity compared with those on the control Ti and pDA-coated Ti surfaces. Parivash Davoudi's group immobilized Heparin and VEGF via self-polymerization and deposition of polydopamine (pDA) on polyurethane (PU) nanofibrous scaffolds [21]. The immobilized VEGF promoted the proliferation of human umbilical vein endothelial cells (HUVECs) through enhanced cells adhesion and cell-scaffold interactions. All these reports determined a great advantage of pDA-assisted growth factor immobilization on tissue regeneration.

The materials modified with different growth factors may get various bioactivities. So that exploring the specific bioactivity of each material modified with different growth factors is quite valuable and interesting. In this research, the pDA-assisted immobilization method was originally applied on bonding between PDGF-BB and PLGA fibrous substrate for in vivo wound healing. In this study, we chose the PLGA fibrous film fabricated by electrospinning as the substrate material. It was because that the PLGA has been proven to be a biocompatible and biodegradable material. And the porous mesh structure of nanofibrous substrate fabricated by electrospinning was beneficial for cell adhesion and migration. First, the effects of pDA-coating and PDGF-BB immobilization on the morphology, compositions, and hydrophilicity of substrates were evaluated in details. Second, the wound healing effect of pDA/PLGA/PDGF-BB substrate was assessed in the dorsal skin of SD rats. Last, the cytokine analysis by ELISA method was employed to evaluate the advantages of pDA/PLGA/PDGF-BB substrate on anti-inflammatory, angiogenesis, and cellular proliferation. The aim of this study was to establish a skin implantation with the potential to accelerate wound regeneration.

## Materials and methods

### Materials

PLGA (lactide/glycolide ratio = 75/25, MW = 100 000) was supplied by Changchun Institute of Applied Chemistry, Chinese Academy of Sciences (CIAC, China). Reagents were received from Sigma-Aldrich (St. Louis, USA) unless specified otherwise. The grade of reagents was analytical grade or higher. PDGF-BB was purchased from Peprotech (Trenton, USA).

## Electrospinning of fibers

PLGA was dissolved in chloroform to obtain a 8% (w/v) solution. The polymer solutions were extruded from a 2.5 ml standard syringe attached to a 25G blunted stainless steel needle using a screw extruder at a flow rate of 1 mL/h. A high voltage of 15 kV was applied to drew the polymer solution into fibers on an aluminum foil wrapped collector at a distance of 10 cm from the needle tip. Fibers collected on foil were dried overnight under vacuum and used for the characterization and animal experiments.

## PDA-assisted immobilization of PDGF-BB onto fibers

The pDA-coating process was carried out according to the previous report [22]. The PLGA fibers were soaked in dopamine solution (2 mg/mL in 10 mM Tris·HCl, pH 8.0) and shaken gently on an oscillator for 2 h under 37˚C. In order to remove free dopamine, the pDA-coated fibers were adequately washed with deionized water five times. Afterward, PDGF-BB (100 ng/mL in PBS) was introduced on pDA/PLGA fibers for another 2 h under 4˚C. The substrates were washed with deionized water three times to remove the free PDGF-BB. The PLGA without pDA coating was also set up to adsorb the PDGF-BB as contrast.

In order to demonstrated the stability of the polydopamine coating, a test was conducted. The PLGA and pDA/PLGA fibers were respectively soaked in 3 mL PBS solution (pH 7.2). 100 μL supernatant was collected as sample at each time point and detected the absorbance at 450 nm using an auto microplate reader (Tecan M200, Switzerland) as reported [23]. The quantitative absorption and release tests based on ELISA method were conducted as follow. When PDGF-BB was absorbed on to PLGA and pDA/PLGA fibers as described above, 100 μL supernatant was collected as sample at each time point. After incubation and wash, the fibers were directly soaked into 3 mL PBS (pH 7.2). 100 μL supernatant was collected as sample at each time point. All samples were diluted according to the range of ELISA kit., and detected quantificationally using the PDGF-BB DuoSet ELISA Development kit (R&D system). The absorption rate and release rate were calculated according to following formula:

$$Absorption\ rate\ (\%) = (Initial\ Amount - Sample\ Amount)/Initial\ Amount \times 100\%$$

$$Release\ rate\ (\%) = Sample\ Amount/Absorption\ Amount \times 100\%$$

## Surface characterization

In order to prove the polydopamine coating on the fibers, the appearance photographs of PLGA and pDA/PLGA fibers were taken. The microstructure images of the membranes were obtained by an environmental scanning electron microscope (ESEM, XL30 ESEM-FEG, FEI). Fiber diameter for each sample was measured and recorded using ImageJ. The average fiber diameter was a statistic based on at least 5 photos from each sample. Fourier transform infrared spectroscopy (FT-IR, Perkin Elmer, FTIR-2000) was used to determine the chemical structure of the membranes. X-ray photoelectron spectroscopy (XPS) (Thermo) was employed to detect the protein-modified substrates. The elemental compositions of the modified substrates were obtained. The water contact angle was detected on a Kriss DSA 10 instrument. The shape of the water drop was recorded by a camera and the water contact angle was measured.

## Animals and experimental protocol

All animal studies were conducted in accordance with the principles and procedures outlined in "Regulations for the Administration of Affairs Concerning Lab-oratory Animals", approved by the National Council of China on October 31, 1988, and "The National Regulation of China

for Care and Use of Laboratory Animals", promulgated by the National Science and Technology Commission of China, on November 14, 1988 as Decree No. 2. Protocols were approved by the Committee of Jilin University Institutional Animal Care and Use. Healthy 8-week-old Sprague Dawley (SD) male rats (purchased from Changchun Institute of Biological Production) weighing ~200g were settled in Experimental Animal Center of Jilin University. Adequate supply of food and water was provided. The rats were randomly divided into four groups of four animals each, and anesthetized by i.p. injection of pentobarbital (40 mg/kg). The middle back area was shaved and disinfected with 75% alcohol. Then, two full-thickness orbicular wounds (diameter of 1.5 cm) were created on each side of their dorsum. Then, different substrates were cut into a suitable size and implanted onto the wounds, followed by injection of penicillin sodium for 4 days continuously. The wounds were covered with sterilized double-layer gauze. After surgery, the animals were placed in separate cages and nourished by assigned person. On day 7 postsurgery, the photographs of wounds were taken. The reduction in the size of wounds was determined from macroscopic images and by estimating the wound area using ImageJ software.

## Immunohistochemistry and immunofluorescence

After 7 days, the rats were sacrificed with an i.p. injection of sodium pentobarbital. The skin scaffolds were excised and fixed in 4% formaldehyde for histological analysis. The fixed samples were embedded in paraffin. Paraffin sections were stained with hematoxylin and eosin (H&E, Sigma). For immunohistochemistry, the samples were stained with primary anti-rat CD31 antibody (1:500, Abcam), followed with HRP-labeled secondary antibody (1:2000, Abcam). After that the DAB was used as chromogenic substrate to locate the vessels. For immunofluorescence, the samples were stained with primary anti-rat CD31 antibody (1:500, Abcam), followed with Alexa-Fluor-488-labeled secondary antibody (1:2000, Abcam). Cell nucleus were dyed with 4,6-diamidino-2-phenylindole (DAPI) for 45 s. The graphs were taken and merged on an inverted microscope (Nikon E200).

## Cytokine analysis

The samples were washed with PBS and smashed into powder in liquid nitrogen, extracted with 40mM Tris-HCl buffer solution (pH 8.0) containing 1 mM PMSF. The suspension was sonicated at 200W for 10 min for cell lysis and then centrifuged at 4˚C, 12000 rpm for 10 min. The supernatant was collected and filtered by 0.22 μm membrane filtration. The cytokine level (TNF-$\alpha$, VEGF, TGF-$\beta$) of the wound tissues was quantitatively detected by the ELISA kits (R&D, Shanghai).

**Statistical analysis.** All data would be reported in the form of mean±standard deviation. A statistical analysis was performed using ANOVA, followed by Tukey's Honestly Significant Difference (HSD) for multiple comparisons. The differences were considered statistically significant for $p < 0.05$. Three or four independent replicates of each experiment were conducted.

# Results and discussions

## Surface characterization of modified fibers

As shown in S1 Fig, the pDA/PLGA fibers were dark brown in appearance, while the PLGA fibers were yellowish. It was due to the dark color of dopamine. In order to remove free dopamine, the pDA-coated fibers had been adequately washed with deionized water five times. Therefore, the dopamine was considered to be stability on the fibers. Even though a little of the dopamine would still fall off the materials.

The microtopography of the membranes was displayed on ESEM. According to the images (Fig 1). It was confirmed that the electrospun films were mainly composed by superposed fibers. The surface topography of the substrates was influenced by pDA coating, and a large amount of polymerized DA envelope was found on the pDA-coated or pDA-mediated peptide-immobilized substrates. As Wang's report [22], pDA coating would obviously change the microtopography of the substrate. A large amount of polymerized DA particles was found on the pDA-coated substrate and increased the roughness of material surface. However, according to the result, there was no typical modified DA particles on PLGA fibers. The pDA was spread evenly on the material. It may be ascribed to the nHA in Wang's material. The average fiber diameter for each sample was given in S1 Table. The morphology of fibers on the same sample might be non-uniform to a certain degree. It was because that many factors could influence the morphology of fibers. And the voltage during preparation was sometimes fluctuant slightly. However, there was no significant difference between samples.

FI-TR spectrum of pDA/PLGA/PDGF-BB, p-DA/PLGA, PLGA/PDGF-BB, and PLGA fibers was shown in Fig 2. The pDA/PLGA/PDGF-BB, pDA/PLGA, and PLGA/PDGF-BB nanofibrous films showed the characteristic amide I and amide II peaks of dopamine and

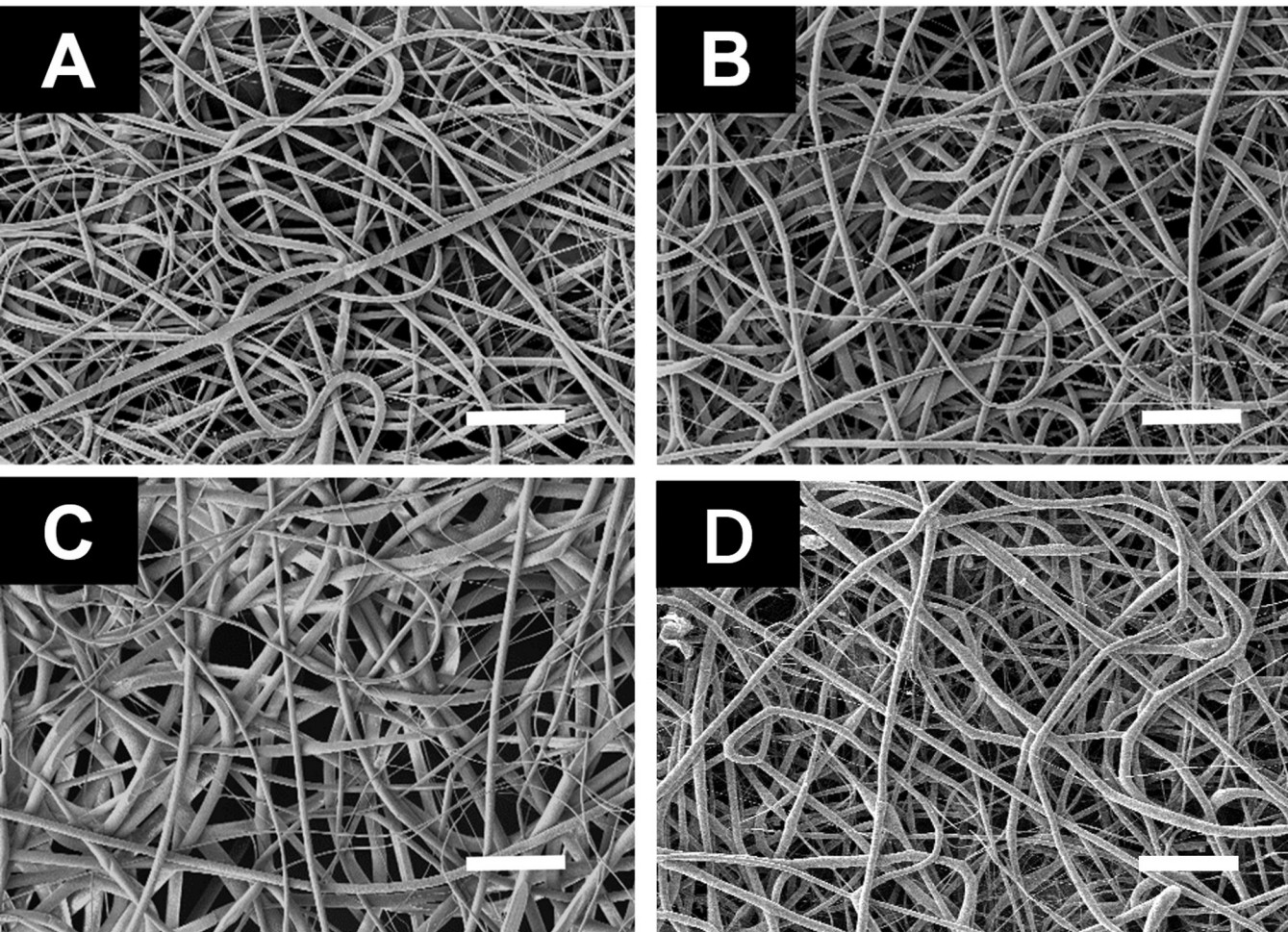

**Fig 1. Scanning electron microscope (SEM) photo.** The scanning electron microscope (SEM) photo of the PLGA fiber substrates. (A) PLGA, (B) PLGA/PDGF-BB, (C) pDA/PLGA (D) pDA/PLGA/PDGF-BB. bar = 10 μm.

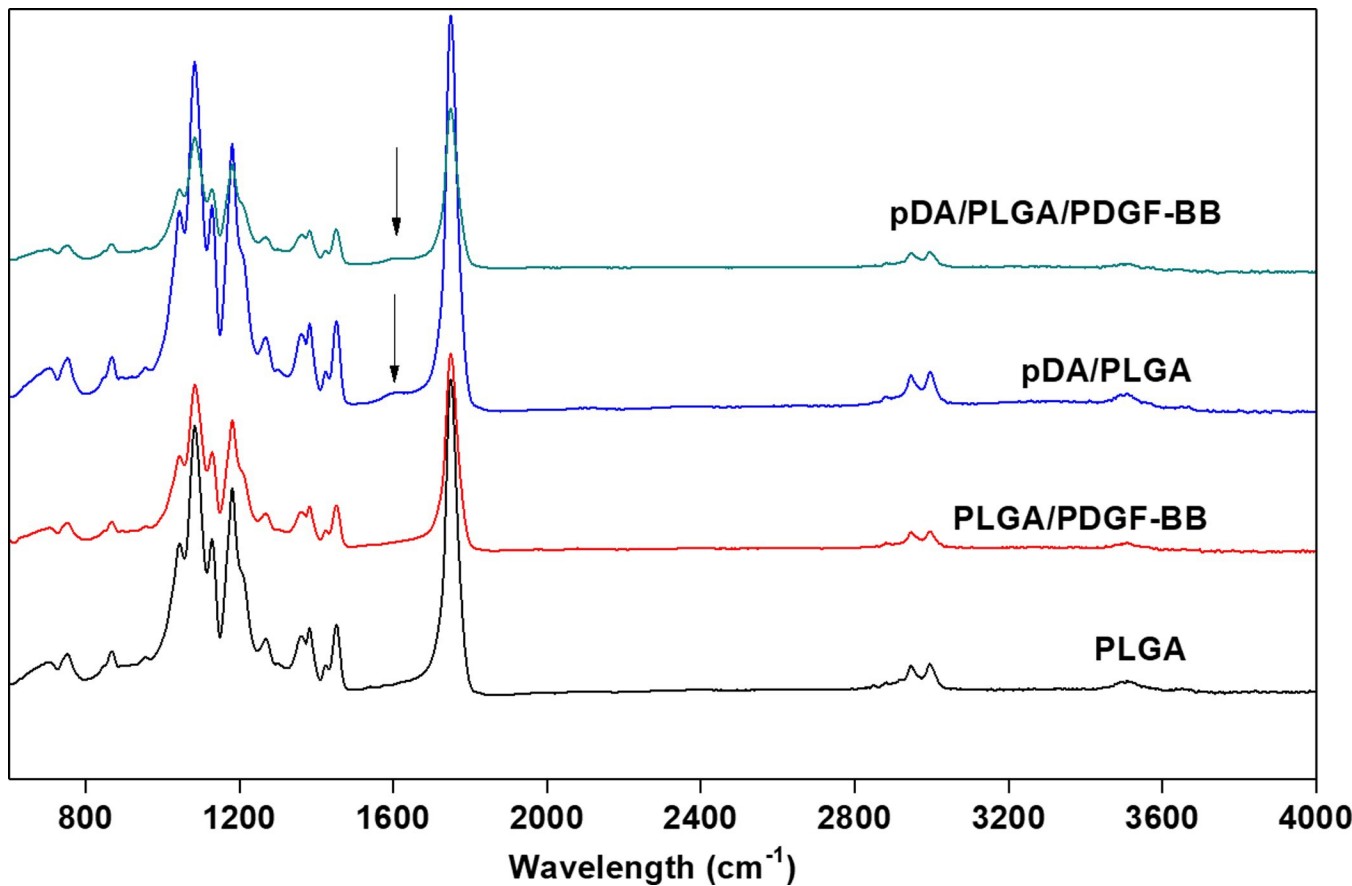

**Fig 2. FI-TR spectrum.** FI-TR spectrum of pDA/PLGA/PDGF-BB, p-DA/PLGA, PLGA/PDGF-BB, and PLGA fibers.

PDGF-BB at 1596 cm-1 in addition to the spectrum of PLGA. Besides that, no significant exclusive peek was displayed.

The result of XPS in powerfully verified the results of FI-TR. Superposed peeks composed with the characteristic peeks were found at 398.3 eV, 399.8 eV and 402.1 eV, which respectively indicated amine (-NH-), imine (= N-),and (N+) in the spectrograms of pDA/PLGA/PDGF-BB, p-DA/PLGA, PLGA/PDGF-BB (Fig 3). Furthermore, it was found that the characteristic peeks on pDA/PLGA/PDGF-BB and p-DA/PLGA were much higher than that on PLGA/PDGF-BB. And the full width at half maximum (FWHM) of the characteristic peek on pDA/PLGA/ PDGF-BB was slightly wider than that on p-DA/PLGA. Table 1 shows the elemental composition before and after peptide coating. The presence of nitrogen element (11.93%) indicated that pDA was successfully coated onto the PLGA nanofibrous films. Incorporation of PDGF-BB via pDA coating further increased the nitrogen content to 15.45%. The chemical deposition of the modified PLGA fibers from FT-IR and XPS analyses demonstrated the efficient immobilization of dopamine and PDGF-BB even after the extensive wash. The assisted PDGF-BB immobilization showed great adsorption advantage in both quantity and stability.

Fig 4 showed the hydrophilia performance of the nanofibrous films modified distinctively. The PLGA nanofibrous films was absolutely hydrophobic and the average contact angle was greater than 120˚. In line with expectation, as the adhesion of different proteins, the hydrophilia of the membranes were more or less improved. However, the groups of PLGA/PDGF-BB was still hydrophobic, while the nanofibrous films of pDA/PLGA/PDGF-BB and pDA/PLGA

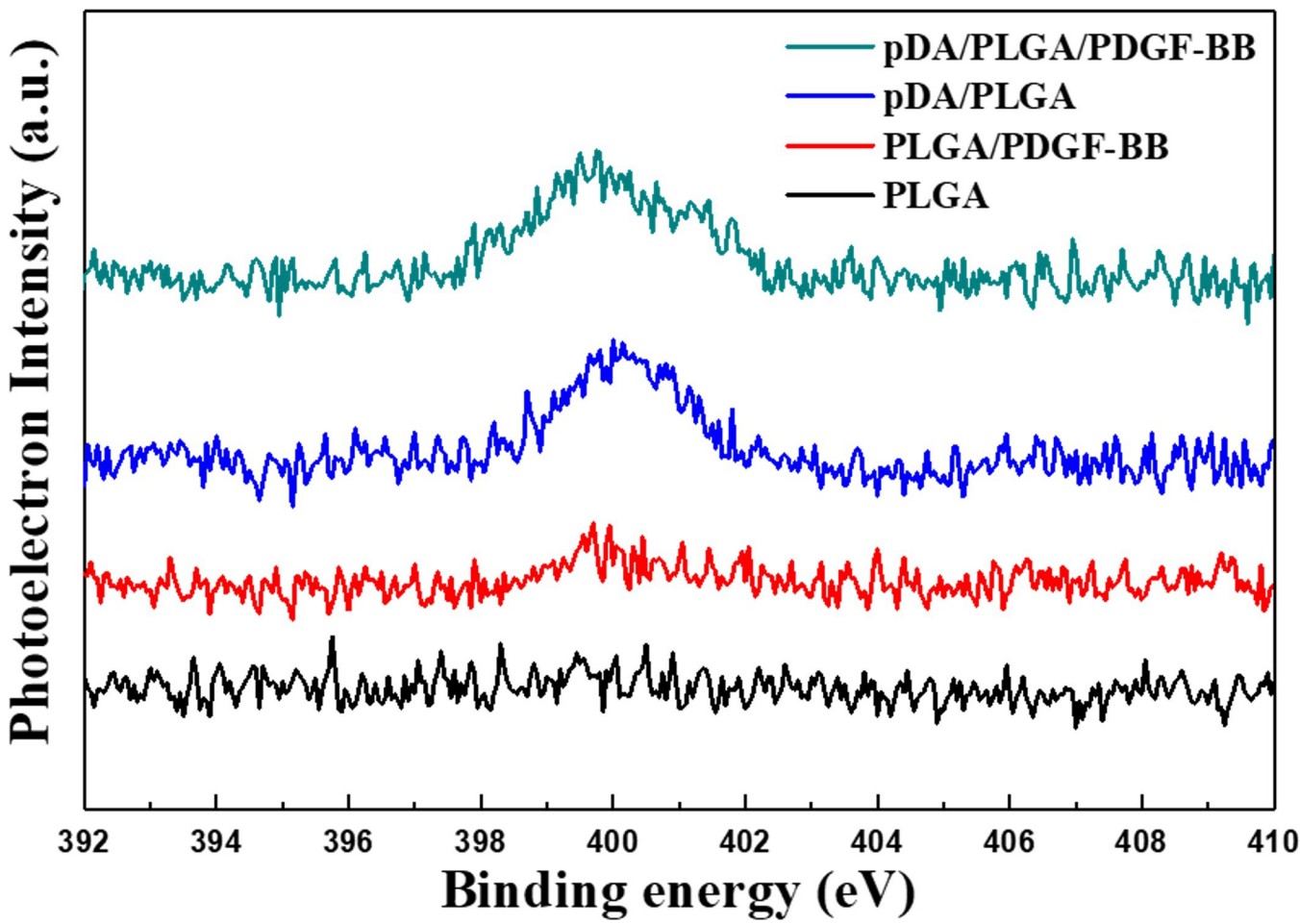

**Fig 3. XPS analyses.** XPS analyses demonstrated the efficient immobilization of PDGF-BB.

performed as a high-hydrophilic material, indicating a significant increase of wettability due to the pDA coating on the substrates.

A polymeric structure with water contact angle less than 55˚ resulted in more cellular adhesion, enhanced proliferation rates, and improved cellular growth regardless of cell type. Parivash Davoudi's study [21] indicated that the introduction of a large number of hydrophilic groups such as–OH,–COOH and–NH2 in the layer of polydopamine could effectively improve hydrophilicity of the PUNF scaffolds. In addition, immobilization of VEGF subsequently further

**Table 1. Surface chemical composition of XPS.** (atomic concentration in %) of substrates analyzed by XPS, $^{**}p < 0.01$.

|  | C | O | N |
|---|---|---|---|
| PLGA | 65.28±0.95 | 29.73±0.98 | 0.11±0.03 |
| PLGA/PDGF-BB | 64.73±1.14 | 28.15±1.21 | 0.18±0.07 |
| pDA/PLGA | 62.12±1.67 | 24.64±1.24 | 11.23±0.43 |
| pDA/PLGA/PDGF-BB | 63.17±1.13 | 21.28±1.51 | 15.45±0.38$^{**}$ |

Surface Chemical Composition (atomic concentration in %) of Substrates Analyzed by XPS,

$^{**}p < 0.01$.

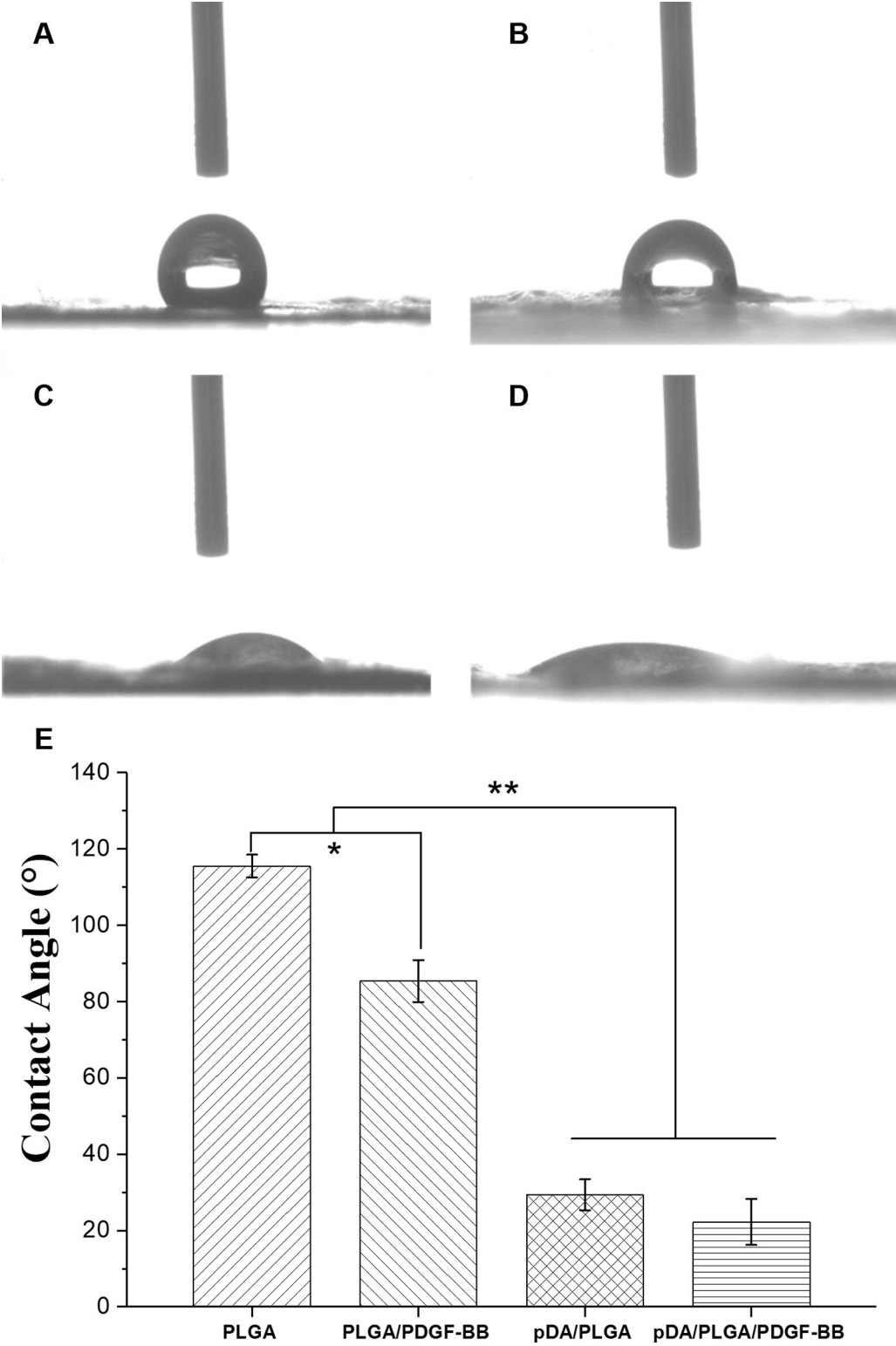

**Fig 4. The hydrophilia performance of the PLGA fiber substrates.** The hydrophilia performance of the PLGA fiber substrates. (A) PLGA, (B) PLGA/PDGF-BB, (C) pDA/PLGA (D) pDA/PLGA/PDGF-BB. (E) Average water contact angle analysis. $^{*}p<0.05$, $^{**}p<0.01$.

decreased the water contact angle. Wang's study [22] also proved that polymerized DA particles were noted on the PLGA film coated with pDA, and the surface wettability were increased. And the surface wettability was extremely important for cellular interaction with materials. A similar phenomenon was also observed in this study. Correspondingly, the wettability of pDA-coated substrates was extraordinarily improved, especially for the pDA/PLGA/PDGF-BB substrate.

### Kinetics of absorption and release

The stability of the coating polydopamine was test. The result was shown in *S2 Fig*. The released dopamine will form nanoparticles in aqueous solution, turning the solution to brown. The color changing can be semiquantitatively detected under OD 450. According to the result, there is no significant difference between PLGA and pDA/PLGA fibers. It indicated that dopamine barely releases from the PLGA fibers. The release of drug is often accompanied by material degradation [24]. However, the situation was different in the polydopamine-assisted system. The coating of polydopamine on materials was formed by the covalent bonds and strong noncovalent forces, including hydrogen bonding, charge transfer, and π-stacking [25]. Polydopamine would usually undergo degradation in vitro in the presence of oxidizing agents such as H2O2. In vivo studies performed by Langer's group have also demonstrated that It took 8 weeks for polydopamine implants to completely degrade [26]. It is quite a long time to wound healing. Therefore, the absorbed PDGF-BB most likely released from fibers due to rupture of hydrogen bond between PDGF-BB and polydopamine.

The kinetics of growth factor absorption and release on PLGA fibers were detected. As shown in Fig 5(A), At each indicated point, the absorption rate of PDGF-BB on pDA/PLGA was significantly higher than that on PLGA ($p < 0.01$), indicating that the retained PDGF-BB on pDA/PLGA was more than that on PLGA. In the in vitro release experiment [Fig 5(B)], sustained release of PDGF-BB on pDA/PLGA and PLGA was followed up to 120 mins. PDGF-BB quickly released from materials at the first 20 mins. After that, the PDGF-BB continued releasing slowly from PLGA fibers, whereas it could be sustained on pDA/PLGA from the 20 minutes to 120 minutes. During 120 minutes, the release rate of LBD-BDNF on on pDA/PLGA was significantly higher than that on PLGA ($p < 0.01$). Therefore, pDA/PLGA fibers could effectively absorb PDGF-BB, keep PDGF-BB retaining on and sustained releasing from substrate in vitro.

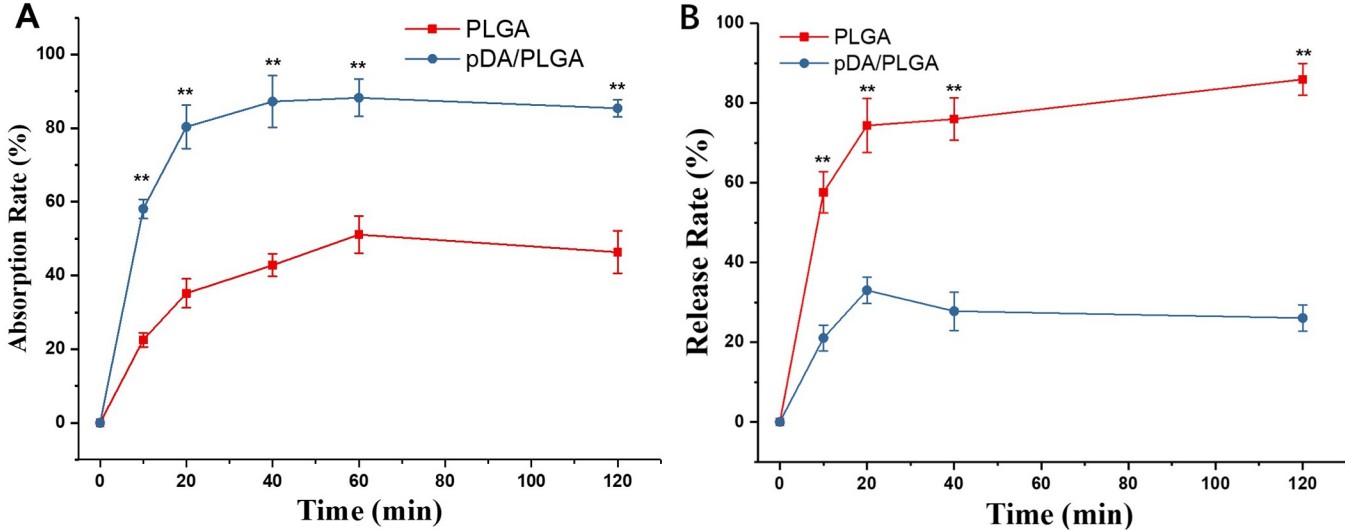

**Fig 5. Kinetics of growth factor absorption and release.** The quantitative (A) absorption and (B) release tests based on ELISA method. $^{**}$p<0.01.

## In *vivo* assessment of pDA/PLGA/PDGF-BB substrate

The wound healing effect of pDA/PLGA/PDGF-BB substrate was assessed in the dorsal skin of SD rats. To measure the wound size, as shown in Fig 6(A), the wound area was photographed every 2 days. The wound area measured each day was normalized by regarding the initial wound area as 100%. The reduction of wound area was calculated by extrapolating from the measured data [Fig 6(B)]. Although the wound size decreased by healing for the PLGA-treated control group, the wound closure process was more accelerated for groups treated with pDA/PLGA/PDGF-BB. The reduction of wound area for the case of pDA/PLGA/PDGF-BB conjugate was over 80%, which was greatly higher than 44.71% for the control group. Remarkably, it was also statistically higher than 55.28% of pDA/PLGA and 51.63% of PLGA/PDGF-BB, reflecting the therapeutic effect of PDGF-BB conjugates on skin wound healing. A combination therapy strategy raised by Robert [16] confirmed that the addition topical PDGF-BB is required to normalize marrow-derived progenitor cells homing/engraftment into the wound to accelerate the wound closure. An earlier research also informed that PDGF augments macrophage-mediated tissue debridement and granulation tissue formation [27]. The immobilization of PDGF prolonged the therapeutic effect, contributed in wound closure.

Histological analysis with H&E staining was performed for further therapeutic analysis (Fig 7). After 7 days, the damaged dermal tissue area was still large for the untreated. Actually, according to the results showed in Fig 6, after treatment with pDA/PLGA/PDGF-BB, the border between proliferative tissues and original tissues was ambiguous, while visible border could be observed in other groups. The borders in groups of PLGA/PDGF-BB and pDA/PLGA were not as clear as that in PLGA group. This result indicated that the PDGF-BB and pDA both could improve fusion between proliferative and original skin. Particularly, according to the result of CD31 staining (Fig 8), a lot of brown positive staining was observed in

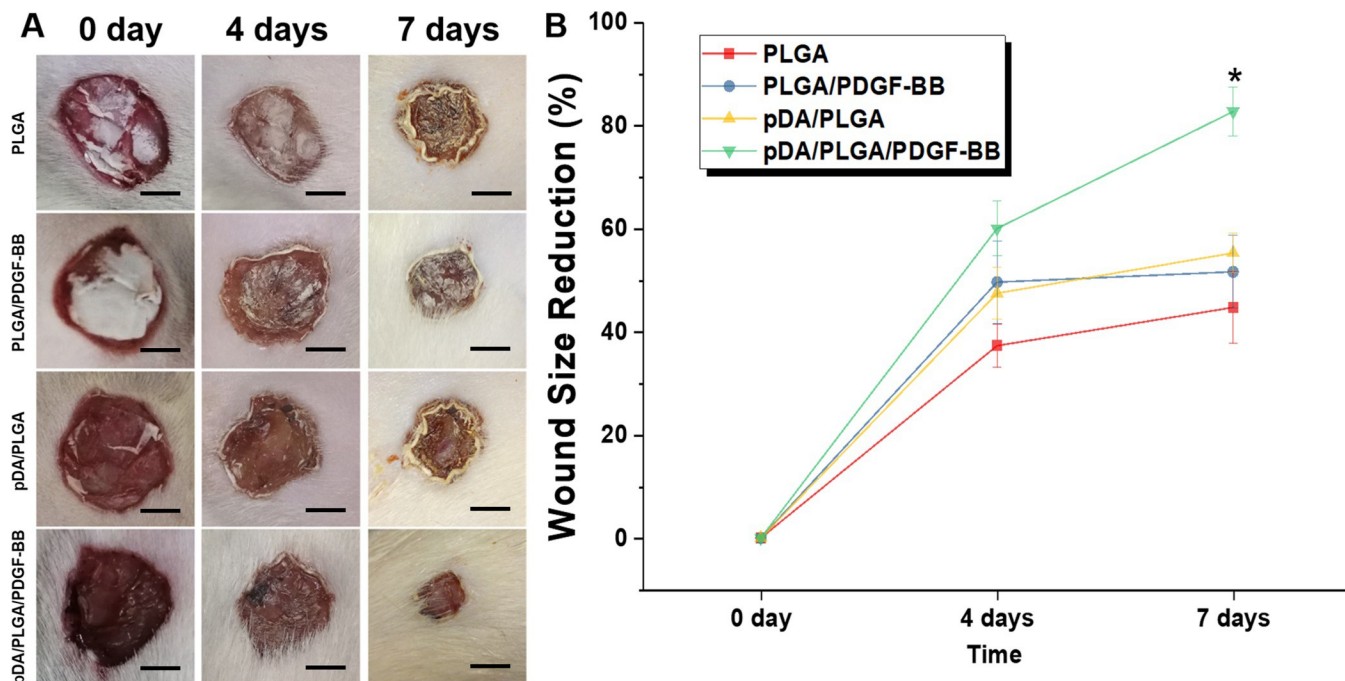

**Fig 6. Macroscopic appearance of wounds.** (A) The macroscopic appearance of the reduction in wound size in different groups of rats treated with various substitutes for 7 days. (B) The wound size reduction (%) with time. Scale bar lengths are 5 mm. These data represent the mean values of three independent experiments ± standard. *$p < 0.05$.

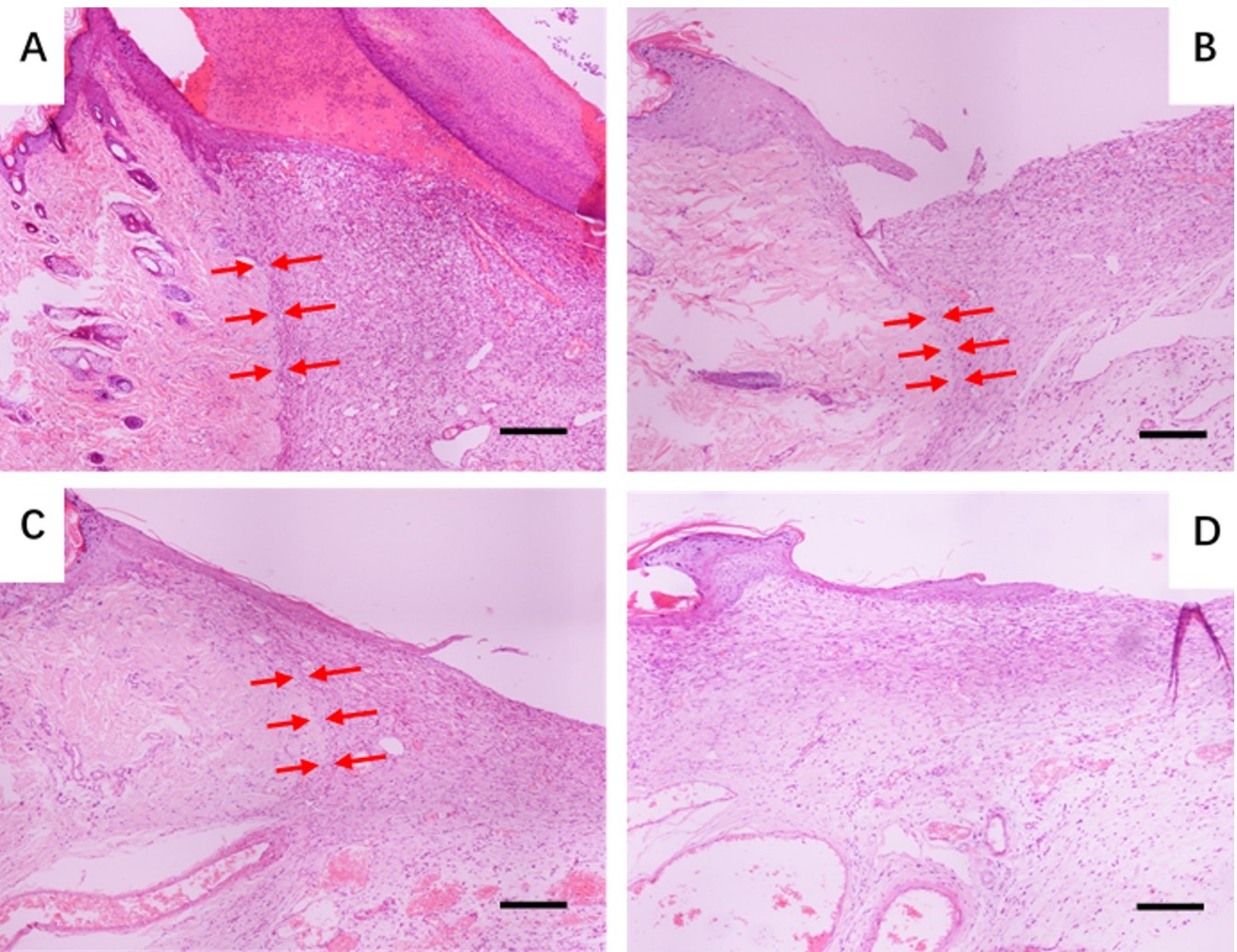

**Fig 7. H&E staining of the tissue.** Histological analysis of the tissue in the group of (A) PLGA, (B) PLGA/PDGF-BB, (C) p-DA/PLGA, and (D) pDA/PLGA/PDGF-BB with H&E staining. (The bars = 200 $\mu$m). The red arrows denote the border between regenerated tissues and normal tissues.

group of pDA/PLGA/PDGF-BB, which was believed to be neovascularization. PDA/PLGA/PDGF-BB substrate treated group showed a highly angiogenesis structure, which might be attributed to the facilitated penetration of PDGF-BB into deep dermal tissues and the prolonged residence of PDGF-BB conjugate. Most of the self-regeneration ability of skin was from basal layer [28, 29], hair follicle [29, 30] and dermis [31]. Therefore, the main task of repairing the full skin injury is to reconstruct the skin structure. It is recognized that extracellular matrix (ECM) deposition and vascularization are the most important in tissue regeneration including skin. PDGF has been accepted to stimulate the proliferation of various cells and the production of ECM [32]. In addition, the PDGF stimulates human fibroblasts to contract floating collagen matrices [33]. In the process of tissue remodeling, PDGF helps to adjust the mitotic activity or the productivity of collagen in dermal fibroblasts [34]. To vascularization, it is worth noting that the effect of PDGF is weaker than that of FGF and VEGF and is not essential for the initial formation of blood vessels [15]. But PDGF is particularly important in blood vessel maturation. The presence of the neovascularization indicated that the vessels were much more mature to take on more blood supply in group of pDA/PLGA/PDGF-BB. In *vivo* experiments have

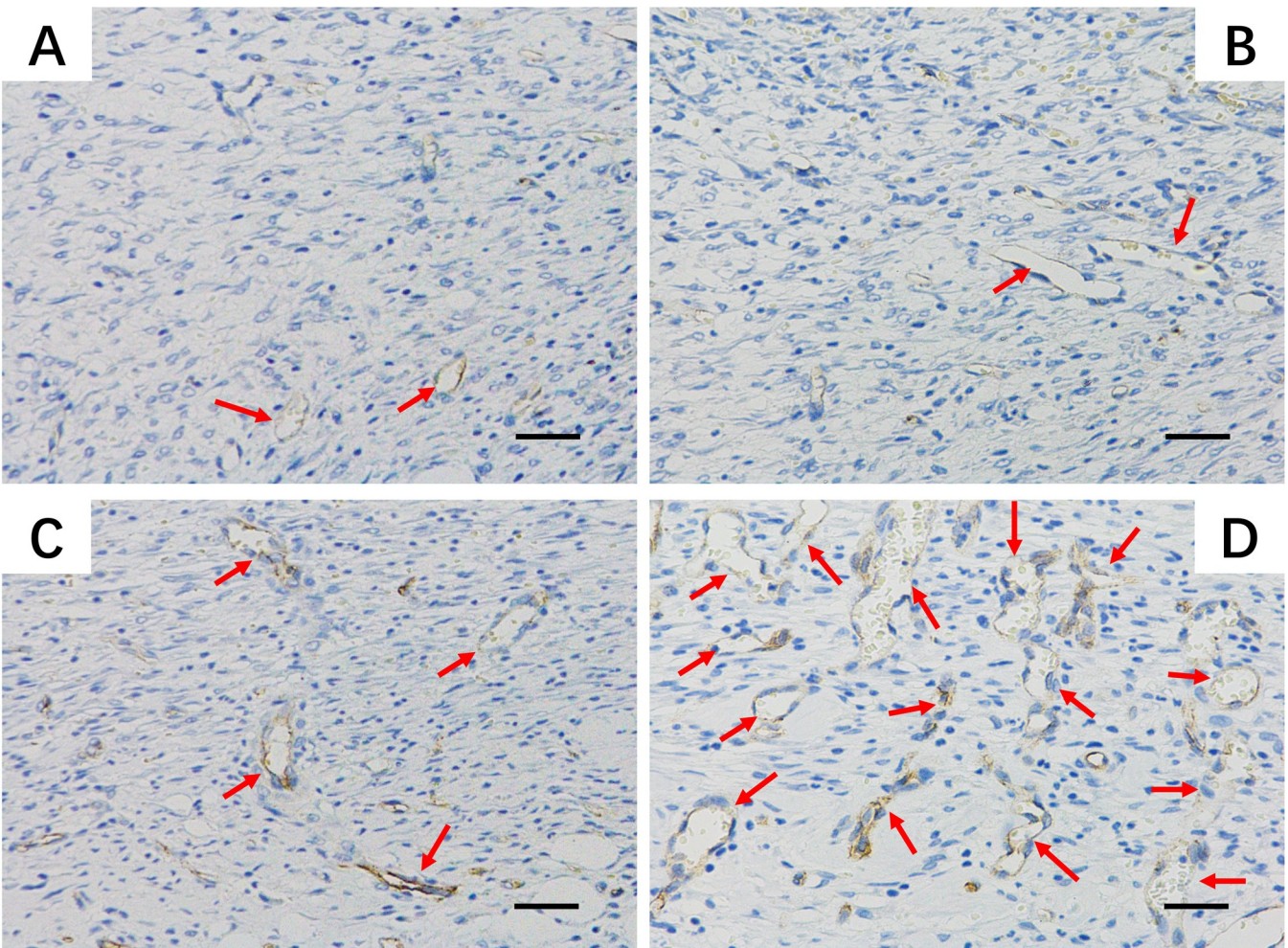

**Fig 8. Immunohistochemical staining of CD31.** Effects of the substitutes on neovascularization of wound area at day 7 after transplantation by examining the expression of CD31. (The bars = 50 $\mu$m).

shown that PDGF is critical for the recruitment of pericytes, thereby increasing the structural integrity of blood vessels [32]. In this study, it was found that the immobilized PDGF-BB greatly enhanced ECM maturation and vascularization, and provide the necessary ECM and vascular structure for skin regeneration.

## Cytokine analysis

The cytokine analysis (Fig 9) in skin tissues also supported the therapeutic effect of pDA/PLGA/PDGF-BB substrate. The pDA/PLGA/PDGF-BB substrate reduced the high level of inflammatory cytokines such as TNF-$\alpha$ induced by wound generation more effectively than the groups of PLGA and PLGA/PDGF-BB ($p<0.05$). Interestingly, injury treated with pDA/PLGA/PDGF-BB showed the increased level of TNF-$\alpha$ than that in pDA/PLGA group. However, the difference was not significant ($p>0.05$). It was reported that the acidic degradation product of PLGA would lead inflammatory reaction in wound. And there was little research supporting the anti-inflammatory role of PDGF-BB in wound area. It was recognized that PDGF could stimulates chemotaxis of neutrophils and macrophages [15]. In other words, PDGF-BB preferred to increase the level of

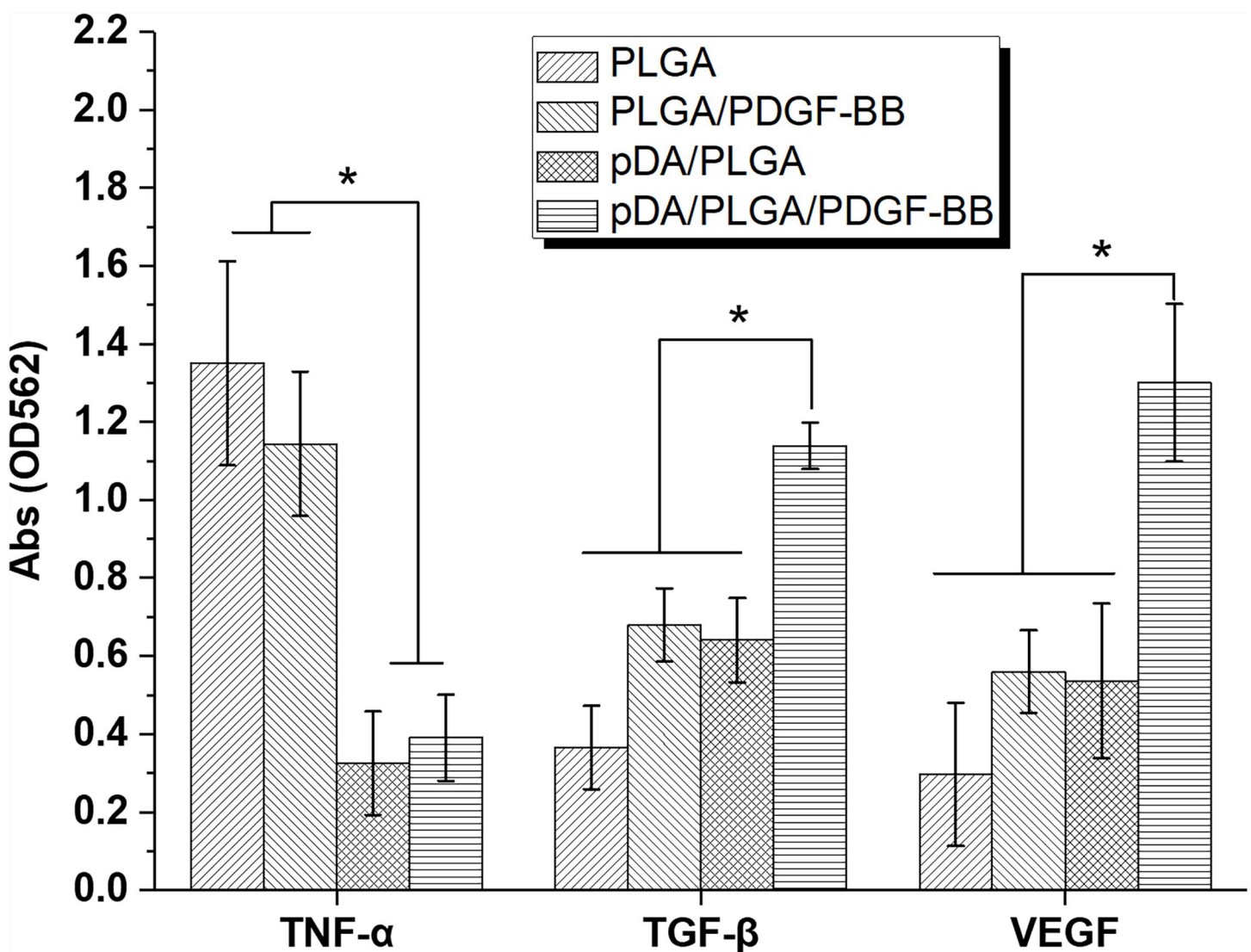

**Fig 9. Cytokine analysis by ELISA method.** Cytokine analysis for TNF-α, TGFβ, and VEGF in the homogenate of dissected wound tissues 7 days after treatment, *$p < 0.05$.

inflammatory cytokines rather than decrease it. The PDGF-BB-recruited immunocytes were available in anti-bacteria and removal of pathological tissue in the early stage of wound regeneration. It was beneficial to reduce the inflammatory reaction in late stage. So that the level of TNF-α in PLGA/PDGF-BB group was a little lower than that in PLGA group. However, the high level of TNF-α was still significantly reduced in pDA/PLGA/PDGF-BB and pDA/PLGA group. It is because that the pDA-coating improved biocompatibility of PLGA [22].

TGF-β could stimulate the cellular proliferation. And VEGF could induce the angiogenesis. In this work, the pDA/PLGA/PDGF-BB substrate also significantly elevated the level of TGF-β and VEGF ($p < 0.05$), believed to benefit the skin tissue regeneration. A previous report has confirmed that PDGF could stimulate macrophages to produce and secrete growth factors such as TGF-β and VEGF [15]. Coincidentally, this phenomenon is verified

in this study. Taken together, we could confirm the feasibility of pDA/PLGA/PDGF-BB substrate from the excellent therapeutic effect on skin wound healing due to the reduction of TNF-α and increasing of TGF-β and VEGF, which was related to the long-term release of the conjugated PDGF-BB.

## Conclusion

In this research, we developed PDGF-BB conjugates incorporated into a PLGA fibers film via pDA-assisted immobilization for skin wound healing by facilitated delivery and prolonged residence of PDGF-BB in wound tissues. The pDA-mediated functionalization allowed for dependable and efficient immobilization of PDGF-BB onto the substrates. Then, the wound healing effect of pDA/PLGA/PDGF-BB substrate was assessed in the dorsal skin of SD rats. With the sustained function on the substrate, PDGF-BB efficiently penetrated into peripheral skin tissues around the wound area with a long residence time against the protease in the wound tissues. The pDA/PLGA/PDGF-BB substrate showed great therapeutic effect on wound healing compared with other control groups via regulating anti-inflammatory, angiogenesis, and cellular proliferation. Absolutely, this report offered an available novel method for skin regeneration.

## Supporting information

**S1 Fig. Appearance of materials.** The appearance photographs of (A) PLGA and (B)pDA/PLGA fibers.
(TIF)

**S2 Fig. Stability of coating polydopamine.** The releasing test of pDA from pDA/PLGA fibers. The PLGA fibers was set as control. All data would be reported in the form of mean±standard deviation.
(TIF)

**S3 Fig. Immunofluorescence staining of CD31.** Effects of the substitutes on neovascularization of wound area at day 7 after transplantation by examining the expression of CD31. (The bars = 50 μm).
(TIF)

**S1 Table. Statistic of fiber diameter of each sample.** Statistic of fiber diameter of each sample. All data would be reported in the form of mean±standard deviation.
(DOCX)

**S1 File.**
(RAR)

## Acknowledgments

We thank Changchun Institute of Applied Chemistry, Chinese Academy of Sciences for its supplement of materials during the research.

## Author Contributions

**Conceptualization:** Xiao Yang, Haitao Fan, Ranwei Li, Ming Zhang.

**Data curation:** Xiao Yang, Yi Zhang.

**Formal analysis:** Peng Zhan, Yi Zhang.

**Funding acquisition:** Ranwei Li, Ming Zhang.

**Investigation:** Xiao Yang, Peng Zhan, Qiushuang Zhang, Yi Zhang.

**Methodology:** Haitao Fan, Ming Zhang.

**Resources:** Xiuyan Wang, Ranwei Li.

**Validation:** Haitao Fan.

**Writing – original draft:** Xiao Yang.

**Writing – review & editing:** Xiao Yang.

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
