## [Decision Letter · Decision Letter 0]

15 Apr 2020

PONE-D-19-33330

Polydopamine-assisted PDGF-BB immobilization on PLGA nanofibrous substrate enhances wound healing via regulating anti-inflammatory and cytokine secretion

PLOS ONE

Dear Dr Zhang,

Thank you for submitting your manuscript to PLOS ONE. After careful consideration, we feel that it has merit but does not fully meet PLOS ONE’s publication criteria as it currently stands. Therefore, we invite you to submit a revised version of the manuscript that addresses the points raised during the review process.

We would appreciate receiving your revised manuscript by May 30 2020 11:59PM. To enhance the reproducibility of your results, we recommend that if applicable you deposit your laboratory protocols in protocols.io, where a protocol can be assigned its own identifier (DOI) such that it can be cited independently in the future. For instructions see: http://journals.plos.org/plosone/s/submission-guidelines#loc-laboratory-protocols

We look forward to receiving your revised manuscript.

Kind regards,

Andrea Caporali, PhD

Academic Editor

PLOS ONE

Journal Requirements:

2. At this time, we request that you  please report additional details in your Methods section regarding animal care, as per our editorial guidelines: 1) Please provide details of animal welfare (e.g., shelter, food, water, environmental enrichment, source of animals) 2) please describe any steps taken to minimize animal suffering and distress, such as by administering analgesics, 3) please include the method of sacrifice and 4) Please describe the post-wound care received by the animals, including the frequency of monitoring and the criteria used to assess animal health and well-being. Thank you for your attention to these requests.

Reviewers' comments:

Reviewer's Responses to Questions

**Comments to the Author**

1. Is the manuscript technically sound, and do the data support the conclusions?

Reviewer #1: Partly

Reviewer #2: No

2. Has the statistical analysis been performed appropriately and rigorously? 

Reviewer #1: Yes

Reviewer #2: Yes

3. Have the authors made all data underlying the findings in their manuscript fully available?

Reviewer #1: Yes

Reviewer #2: No

4. Is the manuscript presented in an intelligible fashion and written in standard English?

Reviewer #1: Yes

Reviewer #2: No

5. Review Comments to the Author

Reviewer #1: It seems a real control untreated group of animals is lacking; Can you explain why?

Figure 6: authors claim that the arrows indicate the border between regenerated tissue and normal tissues. Although this is not clearly visible, wound healing at 7 days should be in the proliferative phase and not in the absence of "regenerated" tissue. "Regenerated" is not a term that can be used in skin wound healing: skin repairs, does not regenerate. Can you clarify?

Figure 7: blood vessels are not visible. The antibody seems to be unspecific, please provide high magnification where endothelial cells are stained.

Reviewer #2: The authors investigate the immobilisation of Platelet-derived growth factor-bb on the surface of PLGA electrospun fibres, aiming at controlled growth factor delivery to stimulate wound healing in full thickness wounds in rats. To deliver on this, a polydopamine coating is attempted on the fibres. My main concerns are around the material design and characterisation, specifically:

1) The data presented do not fully support the formation of a polydopamine coating on the fibres; can the authors carry out some degradation experiments to check that no dopamine leaches out from the fibres? At the moment, the data supports that there is deposition of dopamine species, yet the formation of the polydopamine coating is not fully demonstrated. The authors should provide convincing evidence for this claim.

2) What is the mechanism of growth factor immobilisation and what is the kinetics of growth factor release? The authors should provide data on this using controls made doped with the growth factor in the absence of the dopamine treatment. These data would also help in explaining some of the results obtained with the study in vivo.

3) The SEM images show the internal structure of the fibres. The Authors should assess the average fibre diameter for each sample in order to provide quantitative results. Why do the fibres of the growth factor-loaded scaffolds following dopamine treatment look thicker than all the other sample groups?

4) The authors attribute the increased wound closure of the wounds in receipt of the growth factor loaded and dopamine treated scaffolds to the prolonged PDGF therapeutic effect, due to the effect of the immobilisation. How is this immobilisation achieved and can the authors support this with release data in vitro?

The manuscript also lacks a clear motivation as to why this study is original. The authors state in the introduction that growth factor immobilisation has already been reported. What is the rational for this study then?

6. PLOS authors have the option to publish the peer review history of their article (what does this mean?). If published, this will include your full peer review and any attached files.

Reviewer #1: No

Reviewer #2: No

---

## [Author Response · Author response to Decision Letter 0]

16 May 2020

First of all, we would like to express our great appreciation to the reviewers for their helpful comments and suggestions.

Journal Requirements:

Response: Thank you for your kind comments. The manuscript has been revised to meet PLOS ONE's style requirements.

2. At this time, we request that you please report additional details in your Methods section regarding animal care, as per our editorial guidelines: 1) Please provide details of animal welfare (e.g., shelter, food, water, environmental enrichment, source of animals) 2) please describe any steps taken to minimize animal suffering and distress, such as by administering analgesics, 3) please include the method of sacrifice and 4) Please describe the post-wound care received by the animals, including the frequency of monitoring and the criteria used to assess animal health and well-being. Thank you for your attention to these requests.

Response: Thank you for your kind comments. 

Page 9-10：……All animal studies were conducted in accordance with the principles and procedures outlined in “Regulations for the Administration of Affairs Concerning Lab-oratory Animals”, approved by the National Council of China on October 31, 1988, and “The National Regulation of China for Care and Use of Laboratory Animals”, promulgated by the National Science and Technology Commission of China, on November 14, 1988 as Decree No. 2. Protocols were approved by the Committee of Jilin University Institutional Animal Care and Use. Healthy 8-week-old Sprague Dawley (SD) male rats (purchased from Changchun Institute of Biological Production) weighing ~200g were settled in Experimental Animal Center of Jilin University. Adequate supply of food and water was provided. The rats were randomly divided into four groups of four animals each, and anesthetized by i.p. injection of pentobarbital (40 mg/kg). The middle back area was shaved and disinfected with 75% alcohol. Then, two full-thickness orbicular wounds (diameter of 1.5 cm) were created on each side of their dorsum. Then, different substrates were cut into a suitable size and implanted onto the wounds, followed by injection of penicillin sodium for 4 days continuously. The wounds were covered with sterilized double-layer gauze. After surgery, the animals were placed in separate cages and nourished by assigned person. On day 7 postsurgery, the photographs of wounds were taken. The reduction in the size of wounds was determined from macroscopic images and by estimating the wound area using ImageJ software. ……

Reviewer #1:

1. It seems a real control untreated group of animals is lacking; Can you explain why?

Response: Thank you for your kind comments. The regeneration materials based on PLGA nanofibrous substrate has been widely reported. However, the lack of bioactivity was still a burning problem. This study focused on the enhanced bioactivity of PLGA substrate profited from the polydopamine-assisted PDGF-BB immobilization. An untreated control group was surely useful to show the natural repair of wound. Nevertheless, in order to extrude this emphasis, the PLGA group was set up as control. The results also proved that the ability of PLGA substrate on skin repair was significantly improved by polydopamine-assisted PDGF-BB immobilization. 

2.Figure 6: authors claim that the arrows indicate the border between regenerated tissue and normal tissues. Although this is not clearly visible, wound healing at 7 days should be in the proliferative phase and not in the absence of "regenerated" tissue. "Regenerated" is not a term that can be used in skin wound healing: skin repairs, does not regenerate. Can you clarify?

Response: Thank you for your kind comments. In this part, the main task was to point out the positive effect of polydopamine-assisted PDGF-BB immobilization on fusion between newborn and original skin. Actually, according to the results showed in Figure 6, after treatment with pDA/PLGA/PDGF-BB, the border between proliferative tissues and original tissues was ambiguous, while visible border could be observed in other groups. The borders in groups of PLGA/PDGF-BB and pDA/PLGA were not as clear as that in PLGA group. This result indicated that the PDGF-BB and pDA both could improve fusion between proliferative and original skin. Detailed description of these results was added in the manuscript. However, your comments made me realize that the description was incorrect. Thank you for your correction and we have revised it in the manuscript according to the comments.

Page 13: “……After 7 days, the damaged dermal tissue area was still large for the untreated. Actually, according to the results showed in Figure 6, after treatment with pDA/PLGA/PDGF-BB, the border between proliferative tissues and original tissues was ambiguous, while visible border could be observed in other groups. The borders in groups of PLGA/PDGF-BB and pDA/PLGA were not as clear as that in PLGA group. This result indicated that the PDGF-BB and pDA both could improve fusion between proliferative and original skin. Particularly, according to……”

3.Figure 7: blood vessels are not visible. The antibody seems to be unspecific, please provide high magnification where endothelial cells are stained.

Response: Thank you for your kind comments. Anti-CD31 antibody is commonly used in most histopathology laboratories as a vascular marker and is thus readily available to most histopathologists. During routine practice it became apparent that CD31 was also a good marker of microthrombus in tissue sections. And the microthrombus was quite common in post-mortem tissue and located in some mature vessels. In Figure 7, a lot of granulated positive staining was observed in group of pDA/PLGA/PDGF-BB, which was believed to be microthrombus. These microthrombi were located in the lumen of staining-positive cells. Moreover, the presence of the microthrombi indicated that the vessels were mature enough to take on more blood supply. In order to make this part more clear, the images were updated. And the neovessels were point out with arrows in merged images. In addition, the manuscript was also revised.

 Page 13: ……Particularly, according to the result of CD31 staining (Figure 7), a lot of granulated positive staining was observed in group of pDA/PLGA/PDGF-BB, which was believed to be microthrombus. These microthrombi were located in the lumen of staining-positive cells. PDA/PLGA/PDGF-BB substrate treated group showed a highly angiogenesis structure, which might be attributed to the facilitated penetration of PDGF-BB into deep dermal tissues and the prolonged residence of PDGF-BB conjugate……

Page 14：……But PDGF is particularly important in blood vessel maturation. During routine practice it became apparent that CD31 was also a good marker of microthrombus in tissue sections. And the microthrombus was quite common in post-mortem tissue and located in some mature vessels.(29) The presence of the microthrombi indicated that the vessels were much more mature to take on more blood supply in group of pDA/PLGA/PDGF-BB. In vivo experiments have shown that PDGF is critical for the recruitment……

Reviewer #2: 

The authors investigate the immobilization of Platelet-derived growth factor-bb on the surface of PLGA electrospun fibers, aiming at controlled growth factor delivery to stimulate wound healing in full thickness wounds in rats. To deliver on this, a polydopamine coating is attempted on the fibers. My main concerns are around the material design and characterization, specifically:

1) The data presented do not fully support the formation of a polydopamine coating on the fibers; can the authors carry out some degradation experiments to check that no dopamine leaches out from the fibers? At the moment, the data supports that there is deposition of dopamine species, yet the formation of the polydopamine coating is not fully demonstrated. The authors should provide convincing evidence for this claim.

Response: 

Thank you for your kind comments. In order to prove the polydopamine coating on the fibers, the appearance photographs of PLGA and pDA/PLGA fibers were taken. As shown in Figure S1, the pDA/PLGA fibers were dark brown in appearance, while the PLGA fibers were yellowish. It was due to the dark color of dopamine. In order to remove free dopamine, the pDA-coated nanofibers had been adequately washed with deionized water five times. Therefore, the dopamine was considered to be stability on the fibers. Even though a little of the dopamine would still fall off the materials. 

Page 8: ……In order to prove the polydopamine coating on the fibers, the appearance photographs of PLGA and pDA/PLGA fibers were taken. The microstructure images of the membranes were obtained by an environmental scanning electron microscope (ESEM, XL30 ESEM-FEG, FEI).……

Page 10：……As shown in Figure S1, the pDA/PLGA fibers were dark brown in appearance, while the PLGA fibers were yellowish. It was due to the dark color of dopamine. In order to remove free dopamine, the pDA-coated nanofibers had been adequately washed with deionized water five times. Therefore, the dopamine was considered to be stability on the fibers. Even though a little of the dopamine would still fall off the materials.……

2) What is the mechanism of growth factor immobilization and what is the kinetics of growth factor release? The authors should provide data on this using controls made doped with the growth factor in the absence of the dopamine treatment. These data would also help in explaining some of the results obtained with the study in vivo.

Response: Thank you for your kind comments. the mechanism of growth factor immobilization has been clarified in the part of introduction. Under alkaline conditions, a poly(dopamine) layer is formed on materials through the catechol groups. And the dopamine may offer a lot of activated amino and phenyl to immobilize peptides and proteins through hydrogen-bond interaction. 

In order to clarify the kinetics of growth factor absorption and release on PLGA fibers, the quantitative absorption and release tests based on ELISA method were conducted. Related content was added in revised manuscript. 

Page 8:……The substrates were washed with deionized water three times to remove the free PDGF-BB. The PLGA without pDA coating was also set up to adsorb the PDGF-BB as contrast.

The quantitative absorption and release tests based on ELISA method were conducted as follow. When PDGF-BB was absorbed on to PLGA and pDA/PLGA fibers as described above, 100 μL supernatant was collected as sample at each time point. After incubation and wash, the fibers were directly soaked into 3 mL PBS (pH 7.2). 100 μL supernatant was collected as sample at each time point. All samples were diluted according to the range of ELISA kit., and detected quantificationally using the PDGF-BB DuoSet ELISA Development kit (R&D system). The absorption rate and release rate were calculated according to following formula:

Absorption rate (%) = (Initial Amount-Sample Amount)/Initial Amount × 100%

Release rate (%) = Sample Amount/ Absorption Amount × 100%

……

Page 13: ……

3.2 Kinetics of growth factor absorption and release on PLGA fibers

 The kinetics of growth factor absorption and release on PLGA fibers were detected. As shown in Figure 5(A), At each indicated point, the absorption rate of PDGF-BB on pDA/PLGA was significantly higher than that on PLGA (p < 0.01), indicating that the retained PDGF-BB on pDA/PLGA was more than that on PLGA. In the in vitro release experiment [Figure 5(B)], sustained release of PDGF-BB on pDA/PLGA and PLGA was followed up to 120 mins. PDGF-BB quickly released from materials at the first 20 mins. After that, the PDGF-BB continued releasing slowly from PLGA fibers, whereas it could be sustained on pDA/PLGA from the 20 minutes to 120 minutes. During 120 minutes, the release rate of LBD-BDNF on on pDA/PLGA was significantly higher than that on PLGA (p < 0.01). Therefore, pDA/PLGA fibers could effectively absorb PDGF-BB, keep PDGF-BB retaining on and sustained releasing from substrate in vitro.……

3) The SEM images show the internal structure of the fibers. The Authors should assess the average fiber diameter for each sample in order to provide quantitative results. Why do the fibers of the growth factor-loaded scaffolds following dopamine treatment look thicker than all the other sample groups?

Response: Thank you for your kind comments. In the supporting information of revised manuscript, a table containing the average fiber diameter for each sample was given (Table S1.). The result was a statistic based on at least 5 photos from each sample. However, there was no significant difference between samples. Many factors could influence the morphology of fibers. And the voltage during preparation was sometimes fluctuant slightly. So that the morphology of fibers on the same sample might be non-uniform to a certain degree. The SEM images shown in the original manuscript might be misleading. The image of the growth factor-loaded scaffolds following dopamine treatment has been replaced by another image of the same sample.

Page 8: ……The microstructure images of the membranes were obtained by an environmental scanning electron microscope (ESEM, XL30 ESEM-FEG, FEI). Fiber diameter for each sample was measured and recorded using ImageJ. The average fiber diameter was a statistic based on at least 5 photos from each sample. Fourier transform infrared spectroscopy (FT-IR, Perkin Elmer, FTIR-2000) was used to determine the chemical structure of the membranes.……

Page 11:……It may be ascribed to the nHA in Wang’s material. The average fiber diameter for each sample was given in Table S1. The morphology of fibers on the same sample might be non-uniform to a certain degree. It was because that many factors could influence the morphology of fibers. And the voltage during preparation was sometimes fluctuant slightly. However, there was no significant difference between samples. ……

S1 Table. Statistic of fiber diameter of each sample. All data would be reported in the form of mean±standard deviation. 

 PLGA PLGA/PDGF-BB pDA/PLGA pDA/PLGA/PDGF-BB

Fiber diameter (μm) 1.53±0.64 1.51±0.71 1.61±0.68 1.57±0.74

4) The authors attribute the increased wound closure of the wounds in receipt of the growth factor loaded and dopamine treated scaffolds to the prolonged PDGF therapeutic effect, due to the effect of the immobilization. How is this immobilization achieved and can the authors support this with release data in vitro?

Response: Thank you for your kind comments. the mechanism of growth factor immobilization and the release data in vitro has been given under the comment 2. I hope that the response was clear to answer the questions. 

5）The manuscript also lacks a clear motivation as to why this study is original. The authors state in the introduction that growth factor immobilization has already been reported. What is the rational for this study then?

Response: Thank you for your kind comments. As introduced in manuscript, the Polydopamine-assisted growth factor immobilization has been used in many sections of tissue engineering. The materials modified with different growth factors may get various bioactivities. So that exploring the specific bioactivity of each material modified with different growth factors is quite valuable and interesting. In this research, the pDA-assisted immobilization method was originally applied on bonding between PDGF-BB and PLGA fibrous substrate for in vivo wound healing. Furthermore, this report instructively confirmed the reduction of TNF-α and increasing of TGF-β and VEGF caused by the long-term effect of the conjugated PDGF-BB on pDA/PLGA/PDGF-BB substrate. These results would be helpful to illuminate the underlying mechanism in subsequent studies.

Page 6: ……The materials modified with different growth factors may get various bioactivities. So that exploring the specific bioactivity of each material modified with different growth factors is quite valuable and interesting. In this research, the pDA-assisted immobilization method was originally applied on bonding between PDGF-BB and PLGA fibrous substrate for in vivo wound healing. In this study, we chose the PLGA……

---

## [Decision Letter · Decision Letter 1]

11 Jun 2020

PONE-D-19-33330R1

Polydopamine-assisted PDGF-BB immobilization on PLGA fibrous substrate enhances wound healing via regulating anti-inflammatory and cytokine secretion

PLOS ONE

Dear Dr. Zhang,

Thank you for submitting your manuscript to PLOS ONE. After careful consideration, we feel that it has merit but does not fully meet PLOS ONE’s publication criteria as it currently stands. Therefore, we invite you to submit a revised version of the manuscript that addresses the points raised during the review process.

Specifically, the question on the stability of the coating is  partially answered and it would need an additional degradation test to check the kinetics of the dopamine released. Moreover, CD31 immunohistochemistry in Figure 7 should improved.

We look forward to receiving your revised manuscript.

Kind regards,

Andrea Caporali, PhD

Academic Editor

PLOS ONE

Reviewers' comments:

Reviewer's Responses to Questions

**Comments to the Author**

1. If the authors have adequately addressed your comments raised in a previous round of review and you feel that this manuscript is now acceptable for publication, you may indicate that here to bypass the “Comments to the Author” section, enter your conflict of interest statement in the “Confidential to Editor” section, and submit your "Accept" recommendation.

Reviewer #1: (No Response)

Reviewer #2: All comments have been addressed

2. Is the manuscript technically sound, and do the data support the conclusions?

Reviewer #1: Partly

Reviewer #2: Partly

3. Has the statistical analysis been performed appropriately and rigorously? 

Reviewer #1: Yes

Reviewer #2: Yes

4. Have the authors made all data underlying the findings in their manuscript fully available?

Reviewer #1: Yes

Reviewer #2: Yes

5. Is the manuscript presented in an intelligible fashion and written in standard English?

Reviewer #1: Yes

Reviewer #2: No

6. Review Comments to the Author

Reviewer #1: Dear authors,

thank you for the provided responses. However it is my opinon that green staining that is seen in figure 7 depends on red blood cell unspecific fuorescence.

without high magnification images I cannot say that this is acceptable.

Reviewer #2: The authors have clarified the points raised previously with additional results that support their claims. The question on the stability of the coating is still only partially answered and it would need an additional degradation test to check the kinetics of any dopamine released. I would invite the authors to do that. On the in vivo study, the authors may also discuss the results in the context of the literature, looking e.g. at J. Mater. Chem. B 2016 (4) 7249-7258.

In the introduction the authors should discuss further why this specific PLGA was selected and discuss electrospinning of aliphatic polyesters, e.g. with regards to J. Mech. Behav. Biomed. Mater. 2017 (65) 724-733 and ACS Appl. Bio Mater. 2019 (2) 4258–4270.

7. PLOS authors have the option to publish the peer review history of their article (what does this mean?). If published, this will include your full peer review and any attached files.

Reviewer #1: No

Reviewer #2: No

---

## [Author Response · Author response to Decision Letter 1]

7 Jul 2020

NOTE: The revised contents in the manuscript were labeled in RED. Our responses were written in BLUE just after each comment (BLACK) of the reviewers. The revised content of manuscript was briefly shown following our response.

First of all, we would like to express our great appreciation to the reviewers for their helpful comments and suggestions.

Reviewer #1: Dear authors, thank you for the provided responses. However it is my opinon that green staining that is seen in figure 7 depends on red blood cell unspecific fuorescence. without high magnification images I cannot say that this is acceptable.

Response: Thank you for your kind comments. The images were updated into high magification. And the positions of vascularization and microthrombus were point out with red arrows in merged images. 

Reviewer #2: 

1.The authors have clarified the points raised previously with additional results that support their claims. The question on the stability of the coating is still only partially answered and it would need an additional degradation test to check the kinetics of any dopamine released. I would invite the authors to do that. On the in vivo study, the authors may also discuss the results in the context of the literature, looking e.g. at J. Mater. Chem. B 2016 (4) 7249-7258.

Response: Thank you for your kind comments. The articles you recommended inspired me a lot. The release of drug is often accompanied by material degradation. However, the situation was different in the polydopamine-assisted system. The coating of polydopamine on materials was formed by the covalent bonds and strong noncovalent forces, including hydrogen bonding, charge transfer, and π-stacking. Polydopamine would usually undergo degradation in vitro in the presence of oxidizing agents such as H2O2. In vivo studies performed by Langer’s group have also demonstrated that It took 8 weeks for polydopamine implants to completely degrade. It is quite a long time to wound healing. In order to demonstrate the stability of the coating, a degradation test was conducted as described in manuscript. And the result was shown in Figure S2. 

S2 Fig. Stability of coating polydopamine. The releasing test of pDA from pDA/PLGA fibers. The PLGA fibers was set as control. All data would be reported in the form of mean±standard deviation.

Page 7： ……In order to demonstrated the stability of the polydopamine coating, a test was conducted. The PLGA and pDA/PLGA fibers were respectively soaked in 3 mL PBS solution (pH 7.2). 100 μL supernatant was collected as sample at each time point and detected the absorbance at 450 nm using an auto microplate reader (Tecan M200, Switzerland) as reported.[23]……

Page 13-14：……The stability of the coating polydopamine was test. The result was shown in Figure S2. The released dopamine will form nanoparticles in aqueous solution, turning the solution to brown. The color changing can be semiquantitatively detected under OD 450. According to the result, there is no significant difference between PLGA and pDA/PLGA fibers. It indicated that dopamine barely releases from the PLGA fibers. The release of drug is often accompanied by material degradation.[24] However, the situation was different in the polydopamine-assisted system. The coating of polydopamine on materials was formed by the covalent bonds and strong noncovalent forces, including hydrogen bonding, charge transfer, and π-stacking.[25] Polydopamine would usually undergo degradation in vitro in the presence of oxidizing agents such as H2O2. In vivo studies performed by Langer’s group have also demonstrated that It took 8 weeks for polydopamine implants to completely degrade.[26] It is quite a long time to wound healing. Therefore, the absorbed PDGF-BB most likely released from fibers due to rupture of hydrogen bond between PDGF-BB and polydopamine.……

2. In the introduction the authors should discuss further why this specific PLGA was selected and discuss electrospinning of aliphatic polyesters, e.g. with regards to J. Mech. Behav. Biomed. Mater. 2017 (65) 724-733 and ACS Appl. Bio Mater. 2019 (2) 4258–4270.

Response: Thank you for your kind comments. The recent reports mentioned in the comments about the PLGA and electrospinning was quite interesting. And it has been cited in the revised manuscript to discuss why this specific PLGA was selected and discuss electrospinning of aliphatic polyesters.

Page 3:……Poly[(rac-lactide)-co-glycolide] (PLGA), a FDA-approved polyester, has been widely used in the field of regenerative medicine due to its hydrolytic degradation and good controllability of macro properties. And electrospinning is a simple technique for the efficient production of micro to nano fibers from polymer solutions with high productivity and low cost. It was believed that electrospun fibers had the potential to achieve high drug loading and the ability to sustain drug release. Shih-Feng Chou studied the loading and release of tenofovir (TFV), a hydrophilic small molecule drug, into fibers of polycaprolactone (PCL) and poly (D,L-lactic-co-glycolic) acid (PLGA). The result showed that Sustained release of TFV depended on PLGA content. The PLGA fibers has outstanding sustained-release ability.[8] Amy Contreras’ group encapsulated a photosensitizer (PS) in PCL/PLGA fibers in the PS inert state, so that the antibacterial function would be activated on-demand via a visible light source.[9] This study successfully demonstrates the significant potential of PS-encapsulated electrospun fibers as photodynamically active biomaterial for antibiotic-free infection control. These studies fully demonstrated the advantages of PLGA electrospinning in drug sustained release.……

---

## [Decision Letter · Decision Letter 2]

12 Aug 2020

PONE-D-19-33330R2

Polydopamine-assisted PDGF-BB immobilization on PLGA fibrous substrate enhances wound healing via regulating anti-inflammatory and cytokine secretion

PLOS ONE

Dear Dr. Zhang,

Thank you for submitting your manuscript to PLOS ONE. After careful consideration, we feel that it has merit but does not fully meet PLOS ONE’s publication criteria as it currently stands. Therefore, we invite you to submit a revised version of the manuscript that addresses the points raised during the review process.

The reviewer 1 and I agreed that the CD31 staining in Figure 7 doesn't mark the endothelium; however the green fluorescent staining is due to the auto-fluorescence of erytrocytes in the vessels. We are asking to provide a better staining of the vessels. CD31 antibody can be used together with alpha-SMA to define the vessels. Alternatively, vWF or Isolecti-B4 can be used to better identify the endothelial cells.

We look forward to receiving your revised manuscript.

Kind regards,

Andrea Caporali, PhD

Academic Editor

PLOS ONE

Reviewers' comments:

Reviewer's Responses to Questions

**Comments to the Author**

1. If the authors have adequately addressed your comments raised in a previous round of review and you feel that this manuscript is now acceptable for publication, you may indicate that here to bypass the “Comments to the Author” section, enter your conflict of interest statement in the “Confidential to Editor” section, and submit your "Accept" recommendation.

Reviewer #1: (No Response)

2. Is the manuscript technically sound, and do the data support the conclusions?

Reviewer #1: Yes

3. Has the statistical analysis been performed appropriately and rigorously? 

Reviewer #1: Yes

4. Have the authors made all data underlying the findings in their manuscript fully available?

Reviewer #1: Yes

5. Is the manuscript presented in an intelligible fashion and written in standard English?

Reviewer #1: Yes

6. Review Comments to the Author

Reviewer #1: (No Response)

7. PLOS authors have the option to publish the peer review history of their article (what does this mean?). If published, this will include your full peer review and any attached files.

Reviewer #1: No

---

## [Author Response · Author response to Decision Letter 2]

6 Sep 2020

Reviewer：The reviewer 1 and I agreed that the CD31 staining in Figure 7 doesn't mark the endothelium; however the green fluorescent staining is due to the auto-fluorescence of erytrocytes in the vessels. We are asking to provide a better staining of the vessels. CD31 antibody can be used together with alpha-SMA to define the vessels. Alternatively, vWF or Isolecti-B4 can be used to better identify the endothelial cells.

Response: Thank you for your kind comments. The immunofluorescence images were replaced with immunohistochemical staining of CD31 with high magification. And the previous immunofluorescence images were settled in supporting information. positions of vascularization were point out with red arrows in merged images. 

Page 15-16: Particularly, according to the result of CD31 staining (Fig 8), a lot of brown positive staining was observed in group of pDA/PLGA/PDGF-BB, which was believed to be neovascularization. PDA/PLGA/PDGF-BB substrate treated group showed a highly angiogenesis structure, which might be attributed to the facilitated penetration of PDGF-BB into deep dermal tissues and the prolonged residence of PDGF-BB conjugate. Most of the self-regeneration ability of skin was from basal layer [28, 29], hair follicle [29, 30] and dermis [31]. Therefore, the main task of repairing the full skin injury is to reconstruct the skin structure. It is recognized that extracellular matrix (ECM) deposition and vascularization are the most important in tissue regeneration including skin. PDGF has been accepted to stimulate the proliferation of various cells and the production of ECM [32]. In addition, the PDGF stimulates human ﬁbroblasts to contract ﬂoating collagen matrices [33]. In the process of tissue remodeling, PDGF helps to adjust the mitotic activity or the productivity of collagen in dermal ﬁbroblasts [34]. To vascularization, it is worth noting that the effect of PDGF is weaker than that of FGF and VEGF and is not essential for the initial formation of blood vessels [15]. But PDGF is particularly important in blood vessel maturation. The presence of the neovascularization indicated that the vessels were much more mature to take on more blood supply in group of pDA/PLGA/PDGF-BB. In vivo experiments have shown that PDGF is critical for the recruitment of pericytes, thereby increasing the structural integrity of blood vessels [32]. In this study, it was found that the immobilized PDGF-BB greatly enhanced ECM maturation and vascularization, and provide the necessary ECM and vascular structure for skin regeneration.

---

## [Editor Report · Decision Letter 3]

7 Sep 2020

Polydopamine-assisted PDGF-BB immobilization on PLGA fibrous substrate enhances wound healing via regulating anti-inflammatory and cytokine secretion

PONE-D-19-33330R3

Dear Dr. Zhang,

We’re pleased to inform you that your manuscript has been judged scientifically suitable for publication and will be formally accepted for publication once it meets all outstanding technical requirements.

Kind regards,

Andrea Caporali, PhD

Academic Editor

PLOS ONE
---

## [Editor Report · Acceptance letter]

17 Sep 2020

PONE-D-19-33330R3 

Polydopamine-assisted PDGF-BB immobilization on PLGA fibrous substrate enhances wound healing via regulating anti-inflammatory and cytokine secretion 

Dear Dr. Zhang:

I'm pleased to inform you that your manuscript has been deemed suitable for publication in PLOS ONE. Congratulations! Your manuscript is now with our production department. 

Kind regards, 

on behalf of

Dr Andrea Caporali 

Academic Editor

PLOS ONE